# Women Selling Sex in Russia: Analyzing Women's Appraisal of Exploitation and Mistreatment Using Cognitive Dissonance and Cultural Sex Script Frameworks

Varvara Gulina [1] , Lianne A. Urada [2,3,*], Veronika Odinokova [4] and Maia Rusakova [5]

1    Anschutz Medical Campus, University of Colorado, Denver, CO 80217, USA
2    School of Social Work, San Diego State University, San Diego, CA 92182, USA
3    Department of Medicine & Center on Gender Equity and Health, University of California San Diego, La Jolla, CA 92093, USA
4    NGO Stellit, St. Petersburg 197101, Russia
5    Department of Sociology, St Petersburg University, St. Petersburg 199034, Russia
*    Correspondence: lurada@sdsu.edu

**Abstract:** Globally, over a third of women have experienced physical or sexual violence in their lifetime. In Russia, human trafficking, sexual exploitation, and physical abuse of women are amongst the world's highest. Applying cognitive dissonance theory and sexual script theory, this study explores whether women ($n$ = 654) trading sex in Russia appraise their experiences of entering the commercial sex trade as voluntary or forced. Contributing client factors were also analyzed, including beatings (66%), rape (66%), and humiliation (86%) by clients. Multiple logistic regression assessed whether women who reported voluntarily entering the commercial sex trade were more likely to experience physical abuse but less likely to experience rape (AOR:1.37); were more likely to perceive men using them as decent/caring (AOR = 2.54); were more likely to sell sadistic/masochistic services (AOR: 2.31); and less likely to stop selling sex (AOR: 5.84). Implications of this study reveal the importance of intervention strategies that account for a woman's unawareness of her own exploitation and mistreatment as well as the psychological barriers that prevent her from seeking help. The necessity of recognizing women selling sex as sufferers of coercion and abuse is also emphasized.

**Keywords:** Russian women; sex trafficking; cognitive dissonance; sex scripts; appraisal of exploitation; sexual abuse; sexual exploitation; physical violence; domestic violence; commercial sex trade

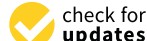



## 1. Introduction

Exploitation of women is a staggering problem that affects one in three women worldwide [1]. Women also experience higher rates of sex trafficking and exploitation globally. The International Labour Office and WFF reported that 4.8 million people were victims of forced sexual labor and coercion in 2016 [2]; this number was mostly made up of females [2]. In 2019, NGOs reported exploitation occurring among females in the thousands [3], while some estimates show the number to be in the millions [4]. Women constitute 71% of all trafficking cases worldwide [5]; some estimates show 80%, with 50% comprising of minors [6]. In Russia, girls and women are at an increased risk of being trafficked and exploited [7,8]. In total, 20,000 to 60,000 females are trafficked every year [9], with many being threatened and intimidated into prostitution by criminal groups [8]. In 2013, over 9000 crimes were related to human trafficking [10]; 35% of those crimes were related to minors [11]. While some attempts have been made to criminalize human trafficking institutionally, Russia is still ranked poorly in its efforts to fully eradicate it [3]. Adolescents who are 15–17 years of age claim to sell and exchange sex for pay [12,13], making Russian women both under and over the legal age at high risk for forced sex [13] and other violence [12–16]. Additionally, women undergo violence and physical assault that is often embedded in cultural manifestations of subordination of women [17]. Internationals studies suggest that women who

sell sex experience more violence from traffickers, such as beatings and torture [6,18], and in Russia, women selling sex become especially vulnerable to client-perpetrated physical violence [14,19] and intimate partner violence [12,14,18].

Even though human trafficking is an issue that permeates nations across the globe [6], Russia poses a higher risk for sexual exploitation in comparison to many other countries [7,10]. Rates of physical violence and exploitation of women in Russia are severe [6,10,12,16], yet the conversation surrounding victimization and coercion of women in Russia does not fall at the forefront of academic discourse and general public knowledge internationally. The paper's objective is to bring awareness about sexual exploitation and physical abuse of women selling sex in Russia by exploring the psychological discrepancy between women's "voluntary" versus "forced" experiences of entering into the commercial sex trade. Cognitive dissonance and cultural sexual script frameworks will be used to explain this discrepancy. Although some general research on cognitive dissonance and sex scripts in relation to women selling sex is available [20–24], the scope is limited. To our knowledge, research on the mentioned theories in relation to perceptions of voluntarily entering the commercial sex trade and appraisals of violence in Russian women who sell sex does not exist.

Cultural sex scripts are social scripts that people use to engage in sexual behaviors that are largely embedded in culture [25]. Internalization of such scripts bleeds into the way individuals perceive sexual interactions and experiences [26], seeing them as either violent or non-violent [27,28]. Women often have very narrow scripts about what a violent sexual scenario looks like [29]. Rape, for example, is often imagined by women as an experience involving strangers and strong acts of violence [30,31], while more complex nuances of rape scenarios are not considered. Due to this narrow interpretation, research has shown that more than half of survivors of rape do not label their experiences as such: a recent meta-analysis found that a majority of women who had experienced the legal definition of rape did not label themselves as victims [32]. Women are less likely to identify sexual assault when they are familiar with the perpetrator [30], and often do not recognize sexual violence when no explicit physical force is present or if the experience does not fit a clearly prescribed societal definition of rape [30]. Because the link between selling sex and sex trafficking is often blurred [33] and because consent in the sex industry is often manipulated and slowly erased [34], women selling sex may not always appraise violence committed against them accurately. Given that Russia is generally imbedded in a socially traditional and conservative culture, cultural sex scripts may shape the way women appraise their personal agency when starting out in the commercial sex industry.

Cognitive dissonance theory asserts that people experience psychological discomfort when their beliefs and behaviors do not align [35]. Resolving the dissonance state is often carried out by psychologically aligning with the cognition or the behavior in question to retain self-consistency [35]. Research has shown that such dissonance becomes a factor in those whom, when faced with the reality of violence being perpetrated against them, will often reduce psychological discomfort by reframing their experiences as non-abusive or "normal" [30,36–42]. Research has also suggested that if a person's positive self-concept is threatened by negatively perceived cognition about the self (such as being a rape victim), the individual will work harder to reduce the discomfort to re-establish a sense of personal control [30,43]. This is supported by a study conducted on women selling sex in Malaysia, where a negative association was found between participants' self-esteem and cognitive distortions [20]. Furthermore, research shows that women experience higher rates of dissonance than men [44] and that adherence to conservative sexual values results in higher cognitive dissonance surrounding women's sexual experiences [44]. Considering the traditional nature of Russian cultural values and gender socialization [45], Russian women trading sex may be more vulnerable to experiencing cognitive dissonance—possibly becoming more likely to construe selling sex as a voluntary action. To provide clarity of theory in relation to our study, we will be following the 2018 Vaidis and Braun approach to cognitive dissonance [46] using the following terms: inconsistency (trigger) and dissonant

state (dissonance). Using this approach, the paper will illustrate the discrepancy in Russian women's appraisal of entering the commercial sex trade as a "voluntary" decision versus a "forced" decision. Russian women's perceptions of beatings, rape, and selling of sadistic and masochistic sex services will also be explored within this theoretical context.

## 2. Materials and Methods

### 2.1. Participants

Data were collected on women in Saint Petersburg and Orenburg, Russia, by the NGO Stellit organization (non-governmental organization in Russia). Both cities provided variability to the lifestyle and social makeup of their residents. Saint Petersburg is the center of culture, art, and education, making up around 5 million residents. Orenburg—a city bordering Kazakhstan—is much smaller in size and popularity, making up approximately 600,000 residents. The majority of women selling sex in St. Peterburg are doing so on the streets [14]. Orenburg, on the other hand, is characterized by more arranged commercial sex trade, with many more women being part of organized crime [14]. Participants were recruited from brothels, hotels, and a railway station. The sample is composed of women 18 and up (*N* = 859); in total, 83.64% were from Saint Petersburg and 16.46% were from Orenburg. For this paper, participants were divided into binary groups of "voluntary" versus "forced" appraisals of entering the commercial sex trade. Those who answered that they did not recall were dropped, which lowered the sample size from 896 to 654 participants. Data were cleaned and prepared by dichotomizing variables including the forced/voluntary dependent variable. Missing data were dropped if not placed into a category.

The study was approved by the ethical review board of the Sociological Institute of the Russian Academy of Sciences and the Deviance and Social Control Department. All measures were taken to ensure the safety of the women trading sex. Risks were minimized by informing local police departments and commercial sex trade managers about the study in an effort to retain confidentiality. An outreach program—which included HIV counseling and education—was used as part of the study as an invitation to participate to maximize benefits for the participants. Interviewers were trained to pay special attention to the participant's capacity to consent to mitigate any pressures from the managers. The interviewers were employees of the outreach program (psychologists and social workers) and Stellit's staff sociologists. All interviewers were pre-trained in interviewing techniques and protection of participants' rights. After the interview was over, psychologists and social workers assisted women who sought help by creating a safety plan.

### 2.2. Measures and Analysis

The study was the largest sociological examination of the commercial sex trade and the social and health needs of women selling sex in Russia since the collapse of the Soviet Union. The full measures and methods of the larger study are described in previous publications [15]; the survey can be found in the appendix for reference. Participants were informed about the purpose of the study and then completed a 90 min face-to-face survey interview in a private area (a small van). Survey interviews addressed a wide scope of topics for data collection, such as demographic profiles, history of abuse, family upbringing, financial profiles, sexual behaviors, marital history, substance use, and other relevant social experiences. Some items included the participant's current age, number of children, education, and marital status, following questions more related to the history and lifestyle of women in the commercial sex trade. The researchers created and pretested the questionnaire to ensure that the language was appropriate for the target sample. Some questions included "Yes" and "No" and "Agree" and "Disagree" responses, while others included Likert Scale answers that range from "Never" to "4 or more times". For the outcome variable, participants were asked, "In your own words, your first commercial sexual contact was rather voluntary or forced?" to which the participants had the option of replying, "It was rather voluntary" or "It was rather forced". Out of 896 participants, 242 stated that they did not recall an incident or episode that marked them entering the

sex business, which dropped our participants from 896 to 654. The remaining participants responded whether they entered either voluntarily or by force.

Descriptive analyses were conducted based on characteristics related to entering the commercial sex trade as a "voluntary" versus "forced" decision. Bivariate analyses captured risk factors such as "age of first commercial contact" and "age of first sex" associated with voluntary vs. forced variables. Reports of violence and perceptions of abuse (humiliation, abasement of human dignity, beatings, appraisal of clients) were also included, along with childhood history of violence. A multiple logistic regression model was used to assess associations between the likelihood of appraising entering the commercial sex trade as voluntary with other factors (thoughts about leaving the commercial sex trade, subjection to violence, appraisal of clients as decent, and selling sadistic/masochistic services). A forward stepwise approach was used in which each variable significant at $p < 0.10$ at the bivariate level was entered one at a time into the multiple logistic regression model in order of significance. If a variable became statistically insignificant (over $p < 0.05$), it was removed.

### 3. Results

Table 1 shows that out of 654 participants, 51.53% claimed to enter the commercial sex trade voluntarily and 48.47% claimed they entered by force. The youngest participant was 15 years old, with the oldest being over 35 years old (15–24 = 39%; 25–34 = 57; 35 and older = 4%). In total, 88.23% of women reported their first sexual encounter as a child. Out of those who reported entering the commercial sex trade voluntarily, more women reported no presence of coercion, threat, or violence during first copulation (49.08%) as opposed to those who reported entering by force (42.51%). Those who claimed voluntarily entering the commercial sex trade were also more likely to have experienced abuse as children (32.42%) than those who claimed to have entered by force (30.12%).

**Table 1.** Characteristics of women in the commercial sex trade in St. Petersburg and Orenburg, Russia, and their bivariate associations with voluntary vs. forced appraisal of entering the commercial sex trade (*N* = 654).

| | (N = 654) | *p* Value | Total (%) | Forced 317 (48.47) | Voluntary 337 (51.53) |
|---|---|---|---|---|---|
| **Socio Demographics** | | | | | |
| *City* | | 0.00 | | | |
| Saint Petersburg | 547 | | 83.64 | 42.97 | 40.67 |
| Orenburg | 107 | | 16.46 | 8.56 | 7.80 |
| *Age* | | 0.092 | | | |
| Under 15–24 | 234 | | 35.78 | 18.65 | 17.13 |
| 25–34 | 392 | | 59.94 | 30.73 | 29.2 |
| **Risk Factors** | | | | | |
| *Age of First Commercial Contact* | | 0.66 | | | |
| 15–17 | 70 | | 10.7 | 4.59 | 6.12 |
| 18–19 | 125 | | 19.11 | 9.48 | 9.63 |
| 20–25 | 347 | | 53.06 | 30.12 | 22.94 |
| 26 or older | 112 | | 17.13 | 7.34 | 9.79 |
| *Age of first Sex* | | 0.37 | | | |
| Child | 577 | | 88.23 | 42.20 | 46.02 |
| Adult | 77 | | 11.77 | 6.27 | 5.50 |
| **Perceptions Reports of Violence** | | | | | |
| *Less Rape* | | 0.001 | | | |
| Never | 219 | | 33.49 | 15.75 | 17.74 |
| Once | 197 | | 30.12 | 10.40 | 19.72 |
| 2–3 times | 158 | | 24.16 | 14.68 | 9.48 |
| 4 times or more | 80 | | 12.23 | 7.65 | 4.59 |

**Table 1.** *Cont.*

| | (*N* = 654) | *p* Value | Total (%) | Forced 317 (48.47) | Voluntary 337 (51.53) |
|---|---|---|---|---|---|
| *Sustained Beating and Injuries* | | 0.01 | | | |
| Never | 268 | | 40.98 | 22.02 | 18.96 |
| Once | 247 | | 37.77 | 17.58 | 20.18 |
| 2–3 times | 109 | | 16.67 | 7.34 | 9.33 |
| 4 times or more | 30 | | 4.59 | 1.53 | 3.06 |
| *Humiliation and Abasement of Human Dignity* | | 0.001 | | | |
| Never | 86 | | 15.15 | 5.67 | 7.49 |
| Once | 119 | | 18.20 | 7.34 | 10.86 |
| 2–3 times | 221 | | 33.79 | 14.68 | 19.11 |
| 4 times or more | 228 | | 34.86 | 20.80 | 14.07 |
| *Appraisal of men who use their services as decent* | | 0.00 | | | |
| Agree | 561 | | 85.78 | 44.34 | 41.44 |
| Disagree | 93 | | 14.22 | 4.13 | 10.09 |
| *Reason for First Copulation* | | 0.06 | | | |
| Presence of coercion, threat, or violence | 60 | | 9.23 | 5.51 | 3.67 |
| No presence of coercion, threat, or violence | 590 | | 90.77 | 42.51 | 49.08 |
| **Prostitution Factors** | | | | | |
| *Thoughts about giving up Prostitution* | | 0.22 | | | |
| Yes | 570 | | 87.16 | 41.44 | 45.72 |
| No | 84 | | 12.84 | 7.03 | 5.81 |
| *Attempts to quit prostitution* | | 0.44 | | | 0.173 |
| Yes | 334 | | 51.07 | 24.01 | 27.06 |
| No | 320 | | 48.93 | 24.46 | 24.46 |
| *Selling of sadistic/masochistic services* | | 0.00 | | | |
| Yes | 47 | | 7.19 | 5.20 | 1.99 |
| No | 607 | | 92.81 | 43.27 | 49.41 |
| **Childhood History of Violence** | | | | | |
| *Experiences of physical punishment* | | 0.47 | | | |
| Yes | 232 | | 35.47 | 16.51 | 18.96 |
| No | 422 | | 64.53 | 31.96 | 32.57 |
| *Being an object of sexual actions as a child* | | 0.57 | | | |
| Yes | 177 | | 27.06 | 13.61 | 13.46 |
| No | 477 | | 72.94 | 34.86 | 38.07 |
| *How often you were persuaded or coerced of such actions against your will* | | 0.33 | | | |
| Never | 504 | | 77.06 | 37.16 | 39.91 |
| Once | 82 | | 12.54 | 7.19 | 5.35 |
| 2–5 times | 48 | | 7.34 | 3.36 | 3.98 |
| More than 5 times | 20 | | 3.06 | 0.76 | 2.29 |

*N* = 654, out of 896 total for those who could recall whether entering the commercial sex trade was voluntary or forced.

Multiple logistics regression (Table 2) demonstrates that out of 70% of women who were underage when they entered the commercial sex trade, almost 60% claimed that they entered voluntarily. Out of those that claimed they entered the commercial sex trade voluntarily, 66% have reported being raped, 63% sustained beatings and injuries, and 86% underwent humiliation and abasement of human dignity. Those that appraised their

experience of entering the commercial sex trade as voluntary were 2× less likely to report being raped (AOR = 1.37, *p* = 0.001); however, they were 2× more likely to report having sustained beatings and injuries (AOR = 1.65 *p* =0.000) and almost 3× more likely to view men who sexually exploit them as caring/decent (AOR = 2.54, *p* = 0.000). Women who claimed to have voluntarily entered were also less likely to consider leaving the profession (AOR: 5.84). The association between selling sadistic and masochistic services was almost 3× higher for women who appraised their experience of entering the commercial sex trade as voluntary than those who were forced into selling sex (AOR = 2.31, *p* = 0.00).

**Table 2.** Characteristics significantly associated with voluntary or forced appraisal of entering into the commercial sex trade among women selling sex in Saint Petersburg and Orenburg, Russia; multiple logistic regression model (*n* = 654).

| Variable | Adjusted Odds Ratio | Confidence Interval 95% |
|---|---|---|
| **Types of Violence** | | |
| Less rape | 1.37 | 1.13–1.7 |
| Beatings and injuries | 1.65 | 1.32–2.05 |
| Humiliation/Abasement of human dignity | 1.28 | 1.06–1.54 |
| **Selling Sex Characteristics** | | |
| Appraisal of men who use their services as decent | 2.54 | 1.54–4.18 |
| Selling sadistic/maschistic services | 2.31 | 1.16–4.59 |
| Thoughts about giving up prostitution | 5.84 | 1.02–2.73 |

## 4. Discussion

Findings reveal that in our sample, more than half of the Russian women did not see themselves selling sex by force. Almost 60% of women (three out of five) perceived their experiences of entering the commercial sex trade as a voluntary choice, even though 70 percent of women entered under 18 years of age. It is important to consider that much previous research reveals that individuals under 18 cannot fully consent due to an undeveloped brain, power differentials with adults, and psychological limitations [47]. Individuals under 18 years of age in the commercial sex trade are automatically considered child human-trafficking victims [48], yet over half of the girls trafficked in our sample did not perceive themselves as victims of exploitation, sexual coercion, or abuse. This perception may have been primed in the women through internalized self-blame in childhood, as evidenced by the finding that two out of five women did not see their childhood sexual experiences stemming from coercion or persuasion. Due to high rates of experiences of sexual acts and child abuse, it is possible that our sample of women internalized mistreatment as normal from an early age which contributed to the appraisals of entering the sex industry as agentic; further findings support this notion through data that reveal that the more women thought they deserved punishment as children, the more likely they saw entering the commercial sex trade as a voluntary action. Additional attestation to such normalization is indicated by data that the more these women thought their first childhood sexual experience was of their own will, the more likely they perceived themselves entering the commercial sex trade voluntarily. This finding aligns with common and standard research findings that highlight children internalizing blame and taking fault for the abuse inflicted on them by adults [49].

The discrepancy between objective victimhood and perception of appraisal of consent is further shown by a significant negative association between women reporting lower rates of rape but higher male-perpetrated experiences of beatings. This points to the possibility that these women experience psychological inconsistency and reduce dissonance by aligning themselves with the cognitions that they are not victims of rape, while, at the same time, reporting instances of objective violation, violence, and degradation. This finding is supported by the research on women failing to process rape as a self-protective mech-

anism [28,30–32] and studies indicating that underage girls will often blame themselves for the sexual abuse inflicted on them [50]. Russian women in our sample may attempt to reduce cognitive dissonance by conforming to self-blame and by claiming responsibility for the abuse perpetrated against them.

Such perceptions are also tied to how the women appraised their experiences with clients in the commercial sex industry. The more client-perpetrated beatings the women experienced, the less they reported instances of rape. Given that over half of women (67%) experienced rape, the negative association between rape and beatings could be attributed to psychological discrepancies that serve as a defense mechanism for victims in a hostile situation. This finding is also consistent with research that indicates that individuals will often reframe their negative experiences as less harmful in order to retain positive perceptions of the self and distract themselves from uncomfortable cognitions [42,43]. Research shows that women are often psychologically trapped in navigating two conflicting realities of whether a negative sexual experience "was" or "was not" rape [30]; acknowledging rape or battery would mean acknowledging one is a victim and thus powerless and without control [30,31]. Therefore, it is possible that Russian women who sell sex may attempt to resolve such psychological discomfort by aligning with the cognition that entering the commercial sex trade was their "choice", retaining a more positive self-perception.

Culturally traditional sex scripts may have played a role in the women's interpretation of the highly hostile and threatening situations as more optimistic than they actually were. Higher reports of positive perceptions of clients strongly support such appraisal. This discovery is consistent with findings that women who commit to traditional gender roles are more likely to remain in abusive relationships [51,52], and women who adhere to conservative attitudes about gender are more likely to blame themselves for the abuse inflicted on them [53]. Some studies also show that certain hyper-male attitudes about women's roles and traditional gender norms increase the likelihood of male-on-female abuse perpetration [54], heightening women's risk of becoming victims. These findings are on par with a 2019 study, where Russian women selling sex described many men using their services as "intimate partners", with 45 percent of women reporting objective violence from those clients [55].

It is worth noting that women's perceptions of clients as caring and decent could also be indicative of the grooming process many young women undergo in the commercial sex trade industry [56]. Grooming limits the victim's ability to perceive their exploitation and mistreatment clearly, resulting in victims viewing their perpetrators as friends and helpers [56,57]. Prior degree of intimacy with perpetrators accounts for higher appraisals of consent [57,58] contributing to the women not recognizing their experiences as coercive, abusive, or dangerous. Additionally, many young women who sell sex develop Stockholm Syndrome [56], often defending their captors and perceiving their perpetrators as loving as a survival mechanism [56]. All of these invisible forces may play a role in the women's skewed positive appraisals of perpetrators, as well as the low attempts of leaving the commercial sex trade (below 30%) for women in both categories (i.e., forced vs. voluntary).

Finally, the strong likelihood of selling sadistic and masochist services when claiming to enter the industry voluntarily could reflect women's low self-esteem, low refusal assertiveness, and re-enactments of unprocessed trauma responses [59–61], which warrants further study. Studies show that low self-esteem predicts dysfunctional sexual behaviors and early exploitation predicts future re-victimization, accounting for higher risky sexual behaviors in female victims [59] and resulting in severe psychological and physical consequences [59]. High rates of physical violence, rape, and humiliation may increase women's vulnerability towards more direct forms of violence during sex acts which may lead to a cycle of continual victimization. These findings showcase the susceptibility of Russian women selling sex to becoming victims of additional client-perpetrated violence while undergoing exploitation and abuse in the commercial sex industry.

### 4.1. Implications

Research on unacknowledged abuse is still not fully understood [30–33], yet its invisible harm has been shown to be profound [42,62]. Even though Russian women trading sex in Saint Petersburg and Orenburg may not always be aware of coercion and violence, the abuse they experience strongly predicts serious life outcomes. Research has shown that females who are subjected to abuse and who self-blame for the abuse inflicted on them experience high levels of depression, dissonance, psychological disorders, physical ailments, adverse brain effects, alcoholism, and trans-generational trauma/abuse [63–67]. Individuals who blame themselves for the sexual violence are also less able to refuse future unwanted sexual advances and may become more likely to be re-victimized [68,69]. These factors speak to the harrowing long-term negative consequences (physical, psychological, and social) of Russian women who sell sex and may spill over into the larger Russian population. Such findings potentially speak to the psychological state of victims outside of Russia as well; the mismatch between victim perceptions of their experiences and abuse may not fully account for what is happening to many women around the world today. Coercive forces inherent by the very nature of the commercial sex trade industry [56,57] renders many women unaware of the broad negative scope of their experiences and keeps many trapped in the industry for years [6,7]. These findings also highlight a critical indication that exploited women do not always report objectively on the state of their exploitation and mistreatment, and, therefore, researchers may not always receive accurate data in terms of crime statistics and rates of abuse. More research on this discrepancy is necessary to fully understand the victims of exploitation in the commercial sex industry to inform actionable measures from outside influences. In-depth knowledge of the psychological aspects related to commercial sex trade as well as the incremental sexual violence that sex workers undergo can inform critical interventions.

It is important to note, however, that prostitution and the commercial sex trade are not always considered harmful in some academic spheres [70]. Still, evidence suggests that women selling sex experience more negative psychological and physical trauma [6,57,71,72], and many scholars often point to commercial sex trade as being inherently coercive and exploitative [33,34]. Therefore, we frame our study through the prism that commercial sex trade is inherently marked by coercion and exploitation. Given the inherent power differentials between men and women [57,73–75] and the subordination of many females selling sex both under and over 18 [6,12,57], we stress the importance of recognizing these women as sufferers of abuse.

### 4.2. Future Directions

Although trading sex is illegal in Russia (Article 6.11), it remains one of Russia's gravest problems [14]. Victims of exploitation within the commercial sex trade are not offered enough protection [15], and legally, while punishment for obtaining sex with a minor (16–18) is now outlined in the Criminal Code of the Russian Federation (Article 240.1), the language concerning minors engaging in the commercial sex trade remained underdeveloped for many years [76]. In 2012, the Russian Federation updated the Criminal Code regarding child sex exploitation (Articles 240.1, 242.1, 242.2) [77]; however, trafficking rehabilitation programs still do not receive much financial support, with many shelters remaining closed due to a lack of funding [3]. According to non-governmental organizations, many cases of sex trafficking victims go underreported due to victims' fears of facing the justice system or lack of governmental support; some reports indicate child victims being punished for forced criminality [3]. Generally, there is little public sympathy for women trading sex in Russia, with a survey indicating that as many as 41% assign blame to the women/girls [78] and a pessimism about the state's capacity to address trafficking effectively [78]. More in-depth social understanding of the full scope of the problem may be necessary for prevention.

In terms of physical violence, Russian women represent the status of some of the most unprotected victims in the world [79]. Based on research and statistics derived from published court verdicts in Russia, out of roughly 18,000 women, 12,000 women

died at the hands of their partners from 2011 to 2019 [80]. Since many incidences of physical abuse in Russia go unreported, actual numbers are believed to be even higher. Bills that would offer perpetrator prosecution and penalties and offer victim support and protection were all dropped in Russia in 1993, 2012, and 2016, respectively [81]. In 2017, the Russian government decriminalized most forms of domestic violence [82]. Laws proposed to be discussed in Russia are heavily criticized, mainly due to the perception that government involvement would break apart the traditional family [82,83]. Such lack of state legislation keeps violence against women as one of the most critical and urgent yet unsolved issues in Russia today [84], putting Russian women selling sex at an increased risk of violence from their partners and clients. In comparison to Africa, Asia, Europe, and Oceania, Russia is leading in its partner violence numbers above all three, 17% over Africa with a total of 53% [80]. Comparative statistics between Russia and the US show that out of 100,000 women, 2.2 thousand intentional murders happened within the US; 4.1 were committed in Russia [85], making Russia—within the context of international comparability—in an especially urgent need of social awareness and advocacy.

Additionally, implementing easily accessible service programs is critical. With scant education on the nuances of abuse and rape in countries such as Russia, increased efforts in education about violence (including psychological, emotional, and spiritual) are essential for broader change. While there are organizations that serve, advocate for, and educate women who have undergone violence in Russia, more translational and dissemination work is needed to infiltrate such education into general public spheres—especially low-income audiences. Presentations and workshops in schools with an emphasis on de-normalization of female mistreatment is needed to assist vulnerable girls in identifying red flags of grooming and coercion early—educating them on insidious forms of abuse of power. Moreover, violence prevention, interventions, and education should include components of sex script theory [25] and cognitive dissonance theory [35] so that Russian women become more socially aware of factors that heighten their risk of becoming victims of violence. Educational efforts to empower Russian women may also be instrumental in narrowing the link between the lack of personal life options and the inability to leave abusive situations that many Russian women undergo; the implementation of programs that account for the lack of personal control, harsh economic strains, epidemic rates of alcoholism, and child maltreatment [15,79,86] may all aid in their effectiveness.

Unfortunately, grant and research funding has been limited due to an ever-growing contentious political climate between Russia and the US [84]. With both countries being in especially tense opposition today, it is important that a-political approach to research and outreach continues to be collaborated on honestly and openly, communicating an effort to simply serve Russia's victims without any ideological predispositions. Given that women are a highly vulnerable population, prompting continual philanthropic, social-work-based cooperation between the US and Russia is paramount. Considering the complex cultural challenges in relation to violence beyond Russia is also important—more international research on unreported and unrecognized victims would provide concrete data and would be helpful for lawmakers in legislative processes and lobbyists in advocacy work. Likewise, a culturally competent and empathetic approach is vital in integrating national value systems with programs that focus on women's protection and empowerment. Consideration and respect for the population's unique history, customs, and psychological makeup are all a part of successfully designing educational programs that the public will positively respond to. On a state and legislative level, creatively presenting the issue with full acknowledgment and attentiveness to stances on tradition and religion may all aid in passing innovative laws that protect victims.

### 4.3. Limitations

There are several limitations to this study. The data reported in this paper are cross-sectional, collected at a one-time point, and therefore causality cannot be inferred. With the rapid advances in technology, we recognize that the way Russian women selling sex

engage with men has shifted to online methods. Thus, the current data may not fully reflect the realities of how these women interact with men today. Furthermore, it is also necessary to interpret the data with caution as no previously validated measures were used in this analysis and some variables had missing data. The age of women was not collected as a continuous variable and most of the women were very young. However, the participants did not differ significantly in age. Inaccurate recall or desirability bias could have affected participants' responses, possibly skewing the data results. Some methodological problems such as limits of sampling and a lack of a control group could be a factor in the ultimate findings as well. Lastly, we recognize that the data were collected at a particular time period, with women in two distinct cities, and, therefore, the findings may not be generalizable to the entire population of women trading sex in Russia. Considering all of these factors, this research may not have fully captured the entire aspect of the commercial sex trade accurately and as it relates to the current situation in Russia today.

## 5. Conclusions

Violence against women is largely manifested due to the age-long tolerance of female mistreatment, vulnerability of women, and the normalization of male-on-female violence as standard [87]. Still, the extent of violence against women and its impact in countries such as Russia, in particular, is profound. Our study findings demonstrate that the majority of Russian women in our sample who trade sex in Orenburg and Saint Petersburg, Russia, do not see themselves as experiencing force and coercion. They also experience discrepancies in reports of rape and other violence. Continued early education, prevention, and intervention strategies aimed at young girls' empowerment are necessary to mitigate such inaccurate perceptions of mistreatment early in their development. Moreover, integration of cultural competency into the work of social workers, therapists, and first responders serving women selling sex are vital in enhancing global efforts that lend to positive social change.

In instances within the commercial sex trade, coercion takes on an invisible form [88] and is slowly erased through a lack of choices, power, or knowledge [34,56,57,88]. "The line between coercion and consent is deliberately blurred in prostitution" [88]. To assume the consent of women who are in the commercial sex trade is to negate its harm. Cultivating common goals, respectfully advocating for positive policy development, and using culturally empathetic approaches to target the awareness of everyday people may all be necessary to intervene for women who are not aware that they are being exploited and mistreated.

**Author Contributions:** Conceptualization, V.G. and L.A.U.; formal analysis, V.G. and L.A.U.; data curation, M.R. and V.O.; writing—original draft preparation, V.G.; writing—review and editing, V.G., L.A.U. and V.O.; supervision, L.A.U., V.O. and M.R.; project administration, M.R. and V.O. All authors have read and agreed to the published version of the manuscript.

**Funding:** The study was conducted through the efforts and financial support of NGO Stellit (Saint Petersburg), a non-governmental organization fighting against sexual exploitation of adults and children. This paper was partially supported by the National Institute on Drug Abuse K01DA036439 (Urada).

**Institutional Review Board Statement:** The study was conducted according to the guidelines of the Declaration of Helsinki, and approved by the Ethics Committee of The Sociological Institute of the Russian Academy of Sciences, Deviance and Social control, St. Petersburg, 198005 Russia (Special session of Deviance and Social control on 15 May 2007).

**Informed Consent Statement:** Informed consent was obtained from all subjects involved in the study.

**Data Availability Statement:** The de-identified data presented in this study are available on request from the corresponding author. Data is not publicly available due to the vulnerability of this population.

**Acknowledgments:** We thank all study participants for sharing their personal stories.

**Conflicts of Interest:** The authors declare no conflict of interest.

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
