# Peer review of "Women Selling Sex in Russia: Analyzing Women’s Appraisal of Exploitation and Mistreatment Using Cognitive Dissonance and Cultural Sex Script Frameworks"

_sexes, doi:10.3390/sexes3030034_

Round 1
Reviewer 1 Report
I am impressed by the paper and by the methodological approach and I do believe this paper is very interesting to the readers.
I would suggest in minor revision to be introduced on different levels of the paper:
Including a context of the international comparability or underlying the specifics of one case (Russia) and explaining, why would it be worth to look at Russia as specific case.
This should take place on the level of literature and the text structure.
If I could give a tipp on the form, the table with data could be separated with explanations taking place not only before and after but in between. But it is more a question of readability, not the quality of the data.
Reviewer 2 Report
Based on a survey of a self-selection sample of N=654 women selling sex in two Russian cities, authors declare them as unrecognized victims of abuse and sexual exploitation.
While the complex topic of selling sex is relevant for the target journal, the manuscript and underlying study, unfortunately, do not meet basic standards of social scientific work.
- Authors claim to use cognitive dissonance theory but completely ignore the fact that this theory has failed replication attempts, and, hence, needs to be treated and applied with much more nuance and caution. https://www.frontiersin.org/articles/10.3389/fpsyg.2019.01189/full
- The state of research is not represented adequately.
- Authors provide idiosyncratic definitions and claims that are not in line with the international state of research, such as equating any form of selling sex as “sex trafficking”.
- Strong claims are made without any scientific evidence - instead dubios online sources are quoted see e.g. references 8-9, 16-21 that are not appropriate for an academic paper and also not correctly referenced.
- Authors fail to provide precise theory work, instead they randomly cite 53 (!) references in bulk in one sentence: “In cases of rape and intimate partner violence, women will often reframe their negative experiences with those close to them as "normal" in order to avoid the psychological discomfort of admitting to being a victim [73 -126].“
- “Drawing on these frameworks, this paper will illustrate the discrepancy in Russian women's appraisal of entering the sex trade as a "voluntary" decision versus a forced one.It is also hypothesized that women will report less rape while reporting more instances of other abusive behaviors from men who exploit them. Statistical relationships between clients' perceptions and beatings, rape, and humiliation will also be explored. Within the context of the theories mentioned, we purport that Russian female victims of commercial sexual exploitation will perceive themselves as voluntary agents in their own victimization.” -> It remains unclear if the study is confirmatory (hypothesis-testing) or exploratory (hypothesis-building). The individual hypotheses and research questions need to be spelled out and related with theory and state of research.
- The instrument is not explained at all. All measures need to be presented and explained. Also, the full instrument should be provided in the appendix.
- No data availability statement available. All data and analysis scripts should be shared on osf.io
- Obviously, no established measures were used which is a major limitation.
- It is alarming that according to results tables only a small number of simple variables was measured but respondents were interviewed for 90 (!) minutes which is excessively long. It needs to be described what happened here.
- No information is provided about interviewer qualification and interview style.
- No information about informed consent is provided. Did participants consent to the authors’ overriding of their answers and declaring them all as “victims” unable to understand their own situation?
- Ethically, it is highly questionable that the authors put themselves in the position to override their respondents’ answers.
- It is highly speculative that authors claim to know about respondents “subconsciousness” – no theory or methodology used in the study would allow for any such claims.
- No information about data cleaning and data preparation is provided.
- Statistical analyses remain unclear. AORs are reported but the type of adjustment is not explained.
- Results tables are incomplete, Table 2, for example, does not report samples sizes.
- Abstract does not provide information about sample size and type.
- Concepts such as “Sado/maso services” are used without definition and are not in line with technical terms from the literature.
- Throughout the manuscript, authors use overgeneralizations such as “Findings reveal that more than half of women in the sex trade do not see themselves as victims of force, which supports our hypothesis.” The study cannot make any claims about “women in the sex trade”, only about the self-selection sample from two Russian cities.
Reviewer 3 Report
The topic of this manuscript is very interesting. However, the title of the manuscript raises some concerns. It is not clear why authors chose the word victim if women themselves seem to be unaware of the victimization even according to one of the selected frameworks. And this leads to more in-depth remarks. For example, it does not seem the two frameworks here are very useful since it is well known that women in the trade sex will not recognize themselves as victims. I think this manuscript has an added value although not in the way it is written at the moment. The Introduction is very focused on women in Russia (it should focus on women in sex trade and the studied variables in both Russia and other countries) and two frameworks. I think the authors should focus on different elements of those frameworks and link them to the variables they collected and not build the Introduction on general and self-evident elements of the frameworks in order to prove their hypotheses. It is a good example of how the frameworks could be applied to real life situations and this is just the departing point. I would recommend to rewrite the manuscript and shorten the Introduction while focusing not on the frameworks as these are just the background but the linkages between the frameworks and the studied variables.
Round 2
Reviewer 2 Report
Neither the action letter nor the revision could address my basic concerns.
Author Response
Thank you very much. Attached is the revised version with: 1) highlighted text that speaks about why looking at Russia as a specific case is important, as well as sentences that speak about Russia in terms of international comparability, 2) added a sentence to make the statement clearer and moved around/added a few citations in other parts, 3) small grammar/word changes in other areas.Thank you very much. Attached is the revised version with: 1) highlighted text that speaks about why looking at Russia as a specific case is important, as well as sentences that speak about Russia in terms of international comparability, 2) added a sentence to make the statement clearer and moved around/added a few citations in other parts, 3) small grammar/word changes in other areas.Thank you very much. Attached is the revised version with: 1) highlighted text that speaks about why looking at Russia as a specific case is important, as well as sentences that speak about Russia in terms of international comparability, 2) added a sentence to make the statement clearer and moved around/added a few citations in other parts, 3) small grammar/word changes in other areas.